# Recent Advances in Poly(Ionic Liquid)-Based Membranes for CO_2_ Separation

**DOI:** 10.3390/polym15030667

**Published:** 2023-01-28

**Authors:** Gabriel Bernardo, Hugo Gaspar

**Affiliations:** 1LEPABE, Department of Chemical Engineering, University of Porto, 4200-465 Porto, Portugal; 2Associate Laboratory in Chemical Engineering–ALiCE, Faculty of Engineering, University of Porto, Rua Dr. Roberto Frias, 4200-465 Porto, Portugal

**Keywords:** poly(ionic liquid) membranes, CO_2_ separation, flue gas, structural and morphological characterization, small-angle scattering techniques

## Abstract

Poly(ionic liquid)-based membranes have been the subject of intensive research in the last 15 years due to their potential for the separation of CO_2_ from other gases. In this short review, different types of PIL-based membranes for CO_2_ separation are described (neat PIL membranes; PIL-IL composite membranes; PIL-polymer blend membranes; PIL-based block copolymer membranes, and PIL-based mixed matrix membranes), and their state-of-the-art separation results for different gas pairs (CO_2_/N_2_, CO_2_/H_2_, and CO_2_/CH_4_) are presented and discussed. This review article is focused on the most relevant research works performed over the last 5 years, that is, since the year 2017 onwards, in the field of poly(ionic liquid)-based membranes for CO_2_ separation. The micro- and nano-morphological characterization of the membranes is highlighted as a research topic that requires deeper study and understanding. Nowadays there is an array of advanced structural characterization techniques, such as neutron scattering techniques with contrast variation (using selective deuteration), that can be used to probe the micro- and nanostructure of membranes, in length scales ranging from ~1 nm to ~15 μm. Although some of these techniques have been used to study the morphology of PIL-based membranes for electrochemical applications, their use in the study of PIL-based membranes for CO_2_ separation is still unknown.

## 1. Introduction

The anthropogenic emission of greenhouse gases, such as CO_2_, to the atmosphere has been considered responsible for the accelerated global warming of our planet and, consequently, a serious threat to the future of mankind [1]. Most of the anthropogenic carbon dioxide emissions result from the burning of fossil fuels like coal, natural gas, and oil. The Paris United Nations agreement for Climate in 2015 [2] highlighted the importance of reducing CO_2_ emissions from fossil fuel combustion. There are currently several carbon capture and storage (CCS) technologies being developed to control CO_2_ emissions in an efficient and economical way. These major CCS approaches can be applied to pre-combustion [3,4], post-combustion [3,5], and oxyfuel combustion [3,6].

Pre-combustion capture [4] involves CO_2_ removal before combustion, and takes place in three stages: (i) the hydrocarbon fuel (methane, or gasified coal) is converted primarily into H_2_ and CO (synthesis gas); (ii) synthesis gas is converted into CO_2_ and H_2_ rich streams by the water gas shift reaction; and (iii) CO_2_ is separated from H_2_ and then it can be compressed into liquid and transported to a storage site. Post-combustion capture [5] involves separating CO_2_ from the exhaust gases (flue gas) created by burning fossil fuels. The flue gas consists mostly of N_2_ and CO_2_. The post-combustion CO_2_ separation techniques may involve cryogenic separation, amines solvent-based absorption, and membrane separation. Currently, the chemical absorption of CO_2_ by aqueous amine solutions occupies 90% of the market for CO_2_ separation [7]. A major drawback of a typical amine-based CO_2_ absorption system is that it requires being heated up to ~120 °C to release the captured CO_2_ and regenerate the amine solvent. This high temperature regeneration process makes the technology very energy intensive [8]. The oxyfuel combustion [6] consists in burning the fuel with nearly pure oxygen instead of air. However, as the amount of oxygen required in oxyfuel combustion is significantly larger than in pre-combustion, the CO_2_ capture costs are higher.

There is a need to develop technological solutions for carbon capture and storage that are simpler to operate, more environmentally friendly, and more energy efficient, and in this context membrane separation technology is the most promising option. There are several different types of membranes currently being studied for the separation of CO_2_ from different gas streams (CO_2_/N_2_, CO_2_/CH_4_, CO_2_/H_2_), and these include polymeric membranes [9], carbon molecular sieve membranes [10,11,12,13,14], and poly(ionic liquids) (PIL) membranes [15,16,17,18].

Membranes for CO_2_ separation based on poly(ionic liquids) (PIL), also named polymerized ionic liquids, were first reported in 2007 by Bara et al. [19]. PIL membranes are obtained by polymerizing ionic liquid (IL) monomers, or crosslinking polymer chains containing IL-based functional groups, forming a charged macromolecular architecture. PIL membranes combine the advantages of polymers, namely improvement of the mechanical stability and durability of the membrane, and the advantages of ILs, namely the ability to tailor the chemical and physical properties of the membrane. PILs often exhibit a significantly higher CO_2_ uptake capacity than their corresponding IL monomers [20,21]. Different types of PIL-based membranes have been investigated: (i) neat PIL membranes; (ii) PIL-IL blend membranes; (iii) PIL copolymer membranes; and (iv) PIL-IL-inorganic particle mixed matrix membranes. 

Neat PIL membranes, due to their compact, solid nature, typically present low gas diffusivities and permeabilities, and selectivities well below the Robeson Upper Bound. Additionally, they are usually very brittle and difficult to process into free standing films, requiring the use of expensive rigid and porous supports. Despite many studies being performed that addressed different PIL polycation structures and functionalization, it became evident that neat PIL membranes cannot achieve the gas separation performances needed to be technologically competitive. This intrinsic limitation of neat PIL membranes has stimulated the research on other types of PIL-based membranes [15,16,17].

PIL-IL composite membranes are produced by blending PIL with ILs. The addition of free ILs into PIL membranes can increase drastically the CO_2_ permeability of PIL-IL membranes [22], and the membrane properties can be tuned with respect to the IL content to guarantee high CO_2_ permeabilities without compromising the mechanical stability that is provided by the PIL. These membranes are very stable and can withstand large pressure gradients without leaching, because the Coulombic attraction between PIL and free ILs largely outweighs the external pressure [17]. The gas permeation properties of the free ILs largely affect the CO_2_ separation performance of the composite membranes, and therefore the appropriate choice of an IL is crucial in tuning the PIL-IL membrane performance for CO_2_ separation.

The gas transport in poly(ionic liquid) membranes obeys the sorption-diffusion mechanism, and the gas permeation through a membrane involves three consecutive steps: (1) gas molecules dissolve into the membrane at the high-pressure side; (2) gas molecules diffuse through the membrane under the concentration gradient across the membrane; (3) gas molecules desorb at the low-pressure side of the membrane. Therefore, the permeability (P) can be explained as being equal to the product of gas diffusivity (D) and sorption (S) in the membrane, i.e., P = D·S. The membrane ideal permeability selectivity (*α*_*i*/*j*_), permselectivity, is the ratio of permeabilities (*P_i_/P_j_*) of two permeating species (*i* and *j*) and can also be represented as the product of diffusivity selectivity (*D_i_/D_j_*) and sorption selectivity (*S_i_/S_j_*) [23].
(1)$$\alpha_{i/j}=\frac{P_{i}}{P_{j}}=\frac{D_{i}}{D_{j}}\times \frac{S_{i}}{S_{j}}$$

PIL membranes for CO_2_ separation should have both a high permeability to CO_2_ and a low permeability to the other gases X of the component mixture, i.e., they should have a high selectivity to CO_2_/X mixtures. A higher permeability to CO_2_ decreases the area of membrane needed to separate a certain amount of gas mixture, and a higher selectivity produces CO_2_ with higher purity, both contributing to a reduction in the capital cost of the purification process. In 1991, Robeson showed that in polymeric membranes for gas separation there is a trade-off relationship between permeability and selectivity, as selectivity tends to decrease when the permeability increases [24]. In the case of polymeric membranes, this relationship is described by the so-called Robeson upper-bound, which is a straight line with a negative slope in the log-log plot of selectivity versus permeability of the more permeable gas. Later, in 2008, Robeson revised the upper bound for several gas mixtures based on newly available experimental data [25]. More recently, Araújo et al. [26] proposed a new figure of merit, the Robeson Index (*θ*), to characterize the separation performance of membranes. The Robeson Index is the ratio between the actual selectivity value *α*_*i*,*j*_ and the one corresponding to the Robeson upper bound (*α*_*i*,*j*_ (RUB)), both for a given permeability:$$\theta =\frac{\alpha_{i,j}}{\alpha_{i,j\;\left(RUB\right)}}$$

The permeability to CO_2_ in poly(ionic liquid) membranes is usually much higher than the permeability to other gases such as H_2_, N_2_, and CH_4_. These are CO_2_-selective membranes, that can permeate CO_2_ and retain other gases and impurities. Furthermore, the sorption-diffusion gas transport mechanism in PIL-based membranes can be facilitated by a CO_2_-selective transport mechanism, as shown in Figure 1. In this mechanism, complexation reactions between CO_2_ and CO_2_ carriers increase the transport of CO_2_ through the membrane. By contrast, other non-reacting gases (such as H_2_, N_2_, CO, and CH_4_) do not experience such a transport enhancement, being transported mostly by the simple sorption-diffusion mechanism. This results in a significantly improved permeability of the membrane to CO_2_, with high selectivities towards H_2_, N_2_, CO, and CH_4_ [27].

In this article, we review the most relevant studies of PIL-based membranes, targeting their application in the purification of CO_2_ and performed within the last 5 years (from 2017 onwards). As a benchmark for performance improvements, in Figure 2 we compare the performances reported in these last 5 years with those reported in a very comprehensive review by Marrucho et al. [15] published in 2016. The gas separation performances of PIL-based membranes for different gas pairs (CO_2_/N_2_, CO_2_/H_2_, and CO_2_/CH_4_) are also shown in Table 1.

## 2. Recent Studies

### 2.1. Neat PIL Membranes

In 2017, Nikolaeva et al. [28] synthesized three novel PILs based on polyvinylbenzyl chloride (PVBC): P[VBTMA][Tf_2_N], P[VBHEDMA][Tf_2_N], and P[VBMP][Tf_2_N]. CO_2_ sorption experiments have shown that CO_2_ has a 5 times higher solubility selectivity to P[VBHEDMA][Tf_2_N] than to the two other PILs. These three PILs were used to fabricate thin-film membranes (average thickness of 2–6 μm) on top of porous polyimide supports and tested for their CO_2_ removal from synthetic flue gas. Membranes based on P[VBHEDMA][Tf_2_N] exhibited the highest mixed-gas selectivity (*α*(CO_2_/N_2_) = 41), but lowest CO_2_ permeability (P_CO_2__ = 349 barrer). Scanning electron microscopy (SEM) analysis, has shown that the PIL-support interface was sharp and clear in the membranes based on P[VBTMA][Tf_2_N] and P[VBMP][Tf_2_N], but diffuse on membranes based on P[VBHEDMA][Tf_2_N]. A diffuse interface may indicate penetration of the PIL into the pores of the support, which can reduce the flux through the membrane. Importantly, humidity in the feed gas was shown to increase the flux through the membranes. Later, in 2018, the same research group synthesized [29] a cellulose-acetate (CA)-based poly(ionic liquid) by modification of cellulose acetate with pyrrolidinium cations through alkylation of butyl chloride, substituting the OH groups in the polymer backbone, followed by anion exchange to bis(trifluoromethylsulfonyl)imide, P[CA][Tf_2_N]. PIL membranes based on the neat P[CA][Tf_2_N] exhibited a pure gas permeability to CO_2_ (P_CO_2__) of 8.9 barrer and a perm-selectivity *α*(CO_2_/N_2_) of 26.8; this separation performance, however, was worse than that of the original CA membrane (P_CO_2__ = 13.8 and *α*(CO_2_/N_2_) = 39.5). However, under mixed gas conditions, membranes based on P[CA][Tf_2_N] exhibited a higher permeability to CO_2_ than membranes based on the original CA.

More recently, Yin et al. [30] synthesized neat, crosslinked poly(ionic liquids) (PILs) and studied the effects of the PIL Mw, crosslinker type, and the mass ratio of PIL:crosslinker on the gas separation performances of the corresponding membranes. Both the use of an ether-containing crosslinker and an increase in the crosslinker content, improved the CO_2_ solubility and diffusivity, with the best neat PILs membrane, named LP(1:2), exhibiting a P_CO_2__ of 170 barrer and a CO_2_/N_2_ permselectivity of 36. This membrane was fully amorphous, as determined by XRD. Semi-crystalline neat-PIL membranes, named LT, exhibited a much poorer permeability and permselectivity than other membranes.

### 2.2. PIL-IL Composite Membranes

The group of Tomé and Marrucho at the University of Lisbon, in Portugal, has been particularly active and has contributed significantly to this field of PIL-IL composite membranes for CO_2_ separation [31,32,33,34,35,36]. In 2017, Tomé et al. [31] prepared, by solvent casting, four different homogeneous and stable PIL-IL composite membranes based on two pyrrolidinium-based PILs, and two siloxane-functionalized ILs: two membranes of poly([Pyr_11_][NTf_2_]) blended with 40 and 60 wt% of [(SiOSi)C_1_mim][NTf_2_], and two membranes of poly([Pyr_11_][C(CN)_3_]) blended with 40 and 60 wt% of [(SiOSi)C_1_mim][C(CN)_3_]. The permeability results obtained demonstrated that the incorporation of siloxane-based ILs into PILs increases the membrane permeability to CO_2_ (P_CO_2__) as well as the CO_2_/N_2_ permselectivity. However, the CO_2_/CH_4_ permselectivity is not significantly increased. For the same amount of IL, the membranes containing the [NTf_2_]^−^ anion exhibit higher permeabilities to CO_2_ than membranes containing the [C(CN)_3_]^−^ anion. However, membranes containing the [C(CN)_3_]^−^ anion display higher CO_2_/N_2_ permselectivities. No structural or morphological studies of these composite membranes were reported.

Later, 42 PIL-IL composite membrane combinations, prepared by the simple solvent casting technique, were tested by Teodoro et al. [32]. The PIL comprised a pyrrolidinium polycation backbone ([Pyr_11_]^+^) and cyano-functionalized anions ([N(CN)_2_]^−^, [C(CN)_3_]^−^, or [B(CN)_4_]^−^), and also an [NTf2]^−^ anion was tested as a reference. The 5 ILs comprised either an imidazolium ([C_2_mim]^+^) or a pyrrolidinium ([Pyr_14_]^+^) based cation, and the same four anions used in the PILs, where for each system the IL anion was always different from the PIL anion. Among all the combinations tested, only 21 produced membranes that were free standing and macroscopically homogeneous, and these were all tested for their gas separation performances. The performance of four of these membranes (PIL N(CN)_2_–60 IL C(CN)_3_, PIL B(CN)_4_–60 IL C(CN)_3_, PIL C(CN)_3_–40 IL N(CN)_2_, and PIL C(CN)_3_–60 IL B(CN)_4_) were close to, or surpassed, the 2008 Robeson upper bound for CO_2_/N_2_ separation. While the CO_2_ and N_2_ permeability in these PIL-IL membranes was found to be mainly controlled by gas diffusivity, the CO_2_/N_2_ permselectivity was found to be controlled by the gas solubility. The study of the micro- and nano-morphologies of the membranes was not reported. 

Tomé et al. [33] studied the effect of the PIL molecular weight (namely high M_w_ (average 400–500 kDa), medium M_w_ (average 200–350 kDa), and low M_w_ (average < 100 kDa)), on the physical and gas permeation properties of PIL-IL composite membranes based on pyrrolidinium-based PILs, having [C(CN)_3_]^−^ as the counter-anion and different amounts (20, 40, and 60 wt%) of free [C_2_mim][C(CN)_3_] IL. Free standing PIL-IL membranes could only be obtained with high and medium M_w_ PIL. PIL-IL membranes based on the medium M_w_ PIL exhibited higher CO_2_ permeabilities (14.6–542 barrer) than those based on the high M_w_ PIL (8.0–439 barrer). The membrane permeability to CO_2_, and the CO_2_/N_2_ permselectivity, both increased with the addition of IL. The performance value reported for the CO_2_/N_2_ separation with the medium M_w_ membrane with 60 wt% of IL (P_CO_2__ = 542 barrer and *α*(CO_2_/N_2_) = 54.0) is the best reported in these last 5 years for this membrane type. No morphological studies of the membranes have been reported.

Gouveia et al. [34] prepared, by solvent casting, free-standing PIL-IL membranes using two pyrrolidinium-based PILs: poly([Pyr_11_][C(CN)_3_]) and poly([Pyr11][NTf_2_]). The poly([Pyr_11_][C(CN)_3_]) was blended with 40 and 60 wt% of [C_2_mim][C(CN)_3_] IL. The poly([Pyr_11_][NTf_2_]) was mixed with 40 and 60 wt% of [C_4_mpyr][NTf_2_] IL and with 40 wt% of ([C_2_mim][NTf_2_]). The CO_2_ and H_2_ permeabilities were measured at 20 °C and 35 °C under a transmembrane pressure differential of 100 kPa. The membrane with the best separation performance was based on poly([Pyr_11_][C(CN)_3_]) blended with 60 wt% of [C_2_mim][C(CN)_3_] IL, and achieved a P_CO_2__ = 438 barrer and *α*(CO_2_/H_2_) = 15.1 at 20 °C, and a P_CO_2__ = 505 barrer and *α*(CO_2_/H_2_) = 12.5 at 35 °C. The study of the membrane’s morphology was not reported. More recently, the same authors tested [35] the permeability of free-standing membranes similar to these, at five different temperatures (20, 35, 50, 65, and 80 °C) and using a multicomponent gas mixture with composition: 57.1 vol% of H_2_, 40 vol% of CO_2_, and 2.9 vol% of N_2_, at a total feed pressure of 1 bar. The same trend of gas permeabilities P_CO_2__ > P_H_2__ > P_N_2__ was obtained for all the membranes, like in the previous single gas experiments. Also, similarly to the results previously obtained in single gas testing [34], the membrane with the best separation performance was again based on poly([Pyr_11_][C(CN)_3_]) blended with 60 wt% of [C_2_mim][C(CN)_3_] IL. This membrane achieved a P_CO_2__ = 325 barrer and *α*(CO_2_/H_2_) = 11.4 at 35 °C and a feed pressure of 1 bar.

Gouveia et al. [36] synthesized three PILs by an ion exchange reaction between poly(diallyldimethylammonium) chloride and salts with anions bis(fluorosulfonyl)amide ([FSI]^−^), (trifluoromethyl)sulfonyl-N-cyanoamide ([TFSAM]^−^), and (trifluoromethyl)sulfonyl-N-trifluoroacetamide ([TSAC]^−^). Composite PIL-IL membranes were prepared by solvent casting with these PILs and different amounts (20 wt% and 40 wt%) of ionic liquids (ILs) based on a 1-methyl-3-ethylimidazolium ([C_2_mim]^+^) cation and the same anion as the respective PIL ([C_2_mim][FSI], [C_2_mim][TFSAM] and [C_2_mim][TSAC]). The best membrane for CO_2_/H_2_ separation was based on PIL FSI−40 IL FSI, and exhibited P_CO_2__ = 201 barrer and *α*(CO_2_/H_2_) = 8.9. No structural studies were reported.

Four novel anionic poly(IL)-IL composite membranes were synthesized by Kammakakam et al. [37] using photopolymerization. Two types of photopolymerizable methacryloxy-based IL monomers (MIL-CF_3_ and MIL-C_7_H_7_) were synthesized and photopolymerized with two distinct amounts of free IL (0.5 equiv and 1 equiv) containing the same cation ([C_2_mim][Tf_2_N]) and 20 wt% PEGDA cross-linker. The best separation performance for the three different gas separations: CO_2_/H_2_, CO_2_/N_2_, and CO_2_/CH_4_, was attained with the membrane (MIL-C_7_H_7_/PEGDA(20%)/IL(1 equiv.)), namely, P_CO_2__ = 20.4 barrer and *α*(CO_2_/H_2_) ~ 4.1, *α*(CO_2_/N_2_) ~ 87, and *α*(CO_2_/CH_4_) ~ 119. Wide-Angle X-ray Diffraction (WAXD) of the membranes revealed that they were mostly amorphous as no sharp peaks could be observed. The main halo observed in the four membranes was used to determine the interchain spacing using the Bragg equation, and the largest d-spacing of 6.63 Å was obtained for the best performing membrane.

In a continuation of previously reported work, Yin et al. [30] prepared crosslinked PIL-IL composite membranes and studied the effect of the PIL M_w_, crosslinker type, and mass ratio of PIL:crosslinker on the gas separation performance of the resulting membranes. The optimized membrane achieved a P_CO_2__ of 2070 barrer and a CO_2_/N_2_ permselectivity of 24.6.

In 2021, Vijayakumar et al. [38] synthesized the poly(ionic liquid) 1-bromohexyl-1 methylpiperidinium bromide (Br-6-MPRD) grafted poly(2,6 dimethyl 1,4 phenylene oxide) (ILPPO). Then free-standing PIL/IL composite membranes were made by mixing the PIL with different amounts of free Br-6-MPRD (0, 2, 5, and 10 wt%), and these membranes are named (ILPPO/Br-6-MPRD-X, where X = 2, 5, 10). Permeability measurements showed that the permeability to CO_2_ (P_CO_2__) increased significantly from 69.58 barrer in the neat ILPPO membrane, to 907.20 barrer in the PIL/IL composite membrane with 2 wt% of IL. However, further increases in the IL content led to a drastic reduction in the permeability to CO_2_—this drastic reduction was attributed to a space-filling effect. SEM was used to study the cross-section of the membranes and all of them looked very homogeneous with no visible phase domains. 

### 2.3. PIL-Polymer Blend Membranes

In 2017, Zhang et al. [39] synthesized a series of semi-interpenetrating polymer network (semi-IPN) membranes by incorporating linear polyvinyl acetate (PVAc) into a cross-linked poly (ionic liquid) (c-PIL) network. For the preparation of PVAc/c-PIL semi-IPN membranes, films were first cast from a solution containing appropriate amounts of PVAc and PIL, and a cross-linker and photo-initiator. Then the cast films were placed under a UV-lamp to promote the cross-linking, and finally the cross-linked PVAc/c-PIL membranes were dried in a vacuum oven. For reference, similar uncross-linked membranes with linear PIL (l-PIL) were prepared by the same procedure, just lacking the cross-linker and photo-initiator (membranes PVAc/l-PIL). A structural analysis of the cross-section of the membranes was made using SEM (magnification between ×500 and ×10,000). While the pure c-PIL membrane exhibited no visible morphological structure (Figure 3e), a microscale morphology, with an average domain size of 2 μm, of the minor phase was observed in the PVAc/c-PIL membranes with 30 and 60 wt% c-PIL (Figure 3b,d), respectively. Furthermore, an interconnected co-continuous microstructure of two phases was observed in the PVAc/c-PIL membranes with 50 wt% c-PIL (Figure 3c). By contrast, the uncross-linked membranes (PVAc/l-PIL) exhibited a significant macroscopic phase separation for l-PIL contents above 10 wt%. The permeability of the PVAc/c-PIL semi-IPN membranes to CO_2_ (P_CO_2__) and to N_2_ (P_N_2__) was shown to increase with the amount of c-PIL in the membranes. The membrane reached its best permeability (P_CO_2__) of 36.1 barrer and permselectivity (CO_2_/N_2_) of 59.6, when the c-PIL content was 50 wt%. For c-PIL contents above 60 wt% the PVAc/c-PIL semi-IPN membrane was very brittle and could not be tested. The performance of the PVAc/c-PIL-50 semi-IPN membrane was also studied at different temperatures, and the permeability to CO_2_ increased with increasing temperature, while the CO_2_/N_2_ permselectivity decreased. 

### 2.4. PIL-Based Block Copolymer Membranes

Nellepalli et al. [40] synthesized a series of imidazolium-based homo-PILs and copolymer-PILs, having different side chain groups (ethyl, pentyl, benzyl, and naphthyl) at the imidazolium ring. The membrane forming ability of these homo-PILs and copolymer-PILs was tested both in their neat state and when mixed with different amounts of free [C_2_mim][NTf_2_] IL. Among all the tested combinations, only three originated stable and homogeneous free standing solid membranes: (i) poly(ViPenIm)(Sty) NTf_2_ with 10 wt% of IL; (ii) poly(ViBenIm)(Sty) NTf_2_ with 25 wt% of IL; and (iii) poly(ViNapIm)(Sty) NTf_2_ with 30 wt% of IL. These three membranes, with IL contents of 10 wt%, 25 wt%, and 30 wt%, exhibited permeabilities to CO_2_ (P_CO_2__) of 21.6, 16.5 and 24.5 barrer, respectively, and CO_2_/N_2_ selectivities of 31.7, 32.9 and 34.4, respectively. The anomalous variation of P_CO_2__ with the amount of the IL content was hypothesized as being due to specific structural features of the copolymers—however no structural characterization was performed to elucidate this. The selectivities observed were attributed to the much higher solubility of CO_2_ in the membranes relative to N_2_. The difference in the chemical structures of the co-PILs did not significantly affect the performance of the membranes.

In 2021, Wang et al. [41] compared membranes based on the synthesized block copolymer-grafted SiO_2_ particle brush SiO_2_-g-PMMA-b-PIL, with membranes based on the homopolymer SiO_2_-g-PIL. The mechanical and permeability tests demonstrated that by introducing a PMMA segment on a PIL-based block copolymer membrane, the mechanical properties of the membrane can be improved without compromising its gas separation performance. 

Very recently, the block copolymers [NBM-mIM][Tf2N] and [NBM-ImCnmIm] [Tf_2_N]_2_ (n = 4 and 6) were synthesized by Ravula et al. [42], and then they were cast into thin membranes and tested for their permeability to several pure gases (CO_2_, N_2_, CH_4_, and H_2_). However, all the prepared membranes displayed very modest permeabilities to CO_2_. The cross-sections of the membranes were studied by SEM. The membranes were also subjected to WAXD analysis, that revealed their essentially amorphous nature.

### 2.5. PIL-Based Mixed Matrix Membranes (MMM)

PIL-based mixed matrix membranes (MMMs) combine the benefits of both polymeric PILs and inorganic materials, and they have gained a special interest in the research community over the last few years as a strategy to circumvent the performance trade-off shown by polymeric membranes.

Supported MMMs, based on the mixture “curable PIL/SAPO-34 zeolite/ionic liquid [EMIM][Tf_2_N]”, with different weight ratios of each component, were studied and reported by Dunn et al. in 2019 [43]. The cross-sections of the membranes were analyzed by SEM: no visible voids were detected. The best performing membrane was composed of 64 wt% PIL, 16 wt% SAPO-34, and 20 wt% [EMIM][Tf_2_N], and achieved a P_CO_2__ of 47 barrer and a CO_2_/CH_4_ selectivity of 42.

In 2019, Nabais et al. [44] reported the preparation of mixed matrix membranes (MMMs) based on the pyrrolidinium-based PIL Poly[Pyr_11_][Tf_2_N], the IL [C_4_mpyr][Tf_2_N], and with three different concentrations (10, 20, and 30 wt%) of three different and highly CO_2_ selective metal organic frameworks (MOFs), namely MIL-53(Al), Cu_3_(BTC)_2_, and ZIF-8. Overall, nine different MMMs were prepared (three different MOFs, each in three different concentrations), and the gas separation performances of these were compared to a membrane without MOF. The results obtained showed that the permeability of the membranes to CO_2_ increased with the addition of MOFs, particularly for the two higher loadings (20 and 30 wt%), and the selectivity *α*(CO_2_/H_2_) also increased with the addition of MOFs. Among the different types of MOFs, ZIF-8 promoted the highest CO_2_ permeability (97.2 barrer, with 30 wt% loading), and MIL-53 the highest CO_2_/H_2_ permselectivity (13.3, with 30 wt% loading). The nanostructures of these membranes were not studied. 

More recently, MMMs based on 60 wt% of poly([Pyr_11_][Tf_2_N]) and 40 wt% [C_2_mim][BETI] IL were prepared with different loadings of MOF-5 (between 10 and 30 wt% of the total mass) and were tested for their CO_2_/CH_4_ separation performance by Sampaio et al. [45]. The CO_2_ single gas permeabilities of the prepared membranes are shown in Table 1 as a function of the MOF-5 loading. Although P_CO_2__ increased with the loading of MOF-5, all the performances were well below the Robeson upper bound limit. Furthermore, the addition of MOF-5 originated brittle membranes. SEM images of the cross-sections of the membranes revealed dense morphologies and a uniform dispersion of the MOF into the PIL/IL matrix on a micrometer length scale. 

In 2020, Nikolaeva et al. [46] prepared PIL-IL-inorganic particle mixed matrix membranes by blending the PIL poly(diallyldimethyl ammonium) bis(trifluoromethylsulfonyl)imide (P[DADMA][Tf_2_N]) with the IL N-butyl-N-methyl pyrrolidinium bis(trifluoromethylsulfonyl)imide ([Pyrr_14_][Tf_2_N]) and with zinc di-bis(trifluoromethylsulfonyl)imide (Zn[Tf_2_N]_2_). Two different mixed matrix membranes PIL/IL/Zn[Tf2N]_2_ were prepared with weight ratios of 9:6:1 (P9IL6Zn1) and 9:6:9 (P9IL6Zn9), and the gas separation performance of these was compared with the performance of neat PIL membranes (P9IL0Zn0) and composite PIL/IL membranes (P9IL6Zn0). The P9IL6Zn0 membrane achieved the best performance for CO_2_/N_2_ separation. No structural studies were reported.

## 3. Advanced Structural Characterization of PIL-Based Membranes

Structural and morphological studies of PIL-based membranes for CO_2_ separation have been so far very much limited to the use of some relatively common techniques such as SEM [28,29,38,39,42,44,46] and TEM [41], as well as some WAXS [37,42], with their associated limitations. Furthermore, the reports on the use of small angle X-ray scattering (SAXS) are also still very scarce [38].

Electrons and X-rays interact with atomic electrons, meaning that the greatest contrast is obtained between elements with significantly different atomic numbers. Therefore, in PIL-IL composites the contrast between different phases—as observed using electron microscopy or X-ray scattering techniques—may be weak due to the small difference in scattering length density between the different phases containing atoms with similar atomic numbers. In this sense, neutron scattering can be used as a technique complementary to X-ray scattering, because neutrons interact with the atomic nucleus and the neutron−nucleus interaction can be very different between nuclei of similar atomic numbers. Therefore, the X-ray and neutron scattering profiles can be very different and provide complementary information. Furthermore, whereas X-ray scattering of a system typically produces only one scattering profile of the total structure, neutron scattering using selective isotopic substitutions (isotopes just differ in their nucleus) can produce many different neutron scattering profiles that highlight different atoms, leading to a more complete picture of the total structure, this latter approach, where H and D atoms are replaced, being termed selective deuteration.

Contrary to the structure of PILs for CO_2_ separation membranes, the micro and nanostructures of PILs has been previously studied in greater detail for some other applications, such as for electrochemical applications. An array of techniques has been used in those studies, including a combination of molecular dynamics (MD) simulations, X-ray scattering (WAXS and SAXS), and neutron scattering (SANS), besides some other more common techniques such as electron microscopy (SEM and TEM). These studies are briefly reviewed below.

The Paddison group has developed important work on elucidating the nanomorphologies of PILs for electrochemical applications [47,48,49]. In 2016, using atomistic molecular dynamics (MD) simulations this group investigated the structural properties of a homologous series of poly(nalkyl-vinylimidzolium bistrifluoromethylsulfonylimide) poly-(nVim Tf_2_N) [47]. The backbone-to-backbone distance, and the size of the nonpolar nanodomains, were shown to increase with the alkyl chain length at a rate of 1 Å/CH_2_. Excellent agreement was obtained between the MD simulations and the results from the X-ray scattering experiments. The alkyl chain length dependence of backbone-to-backbone distance on the complete homologous series of PIL poly(C_n_Vim Tf_2_N) (*n* = 2–8), was later studied by the same group [48], using extensive atomistic MD simulations. Excellent agreement was once again observed between the atomistic simulations and the experimental X-ray scattering results, and the backbone-to-backbone correlation length was once again shown to increase with the alkyl chain length. In another similar work performed by the same group [49], atomistic simulations revealed a progressive change in the nanoscale morphology of imidazolium-based PILs with increasing alkyl chain length: discrete apolar islands form initially inside the continuous polar network and then grow beyond the percolation threshold, finally forming a bi-continuous nanostructure of polar and apolar domains. These complex 3D networks were considered very important for the ionic conductivity.

Doughty et al. [50], using WAXS in conjugation with DFT calculations, have shown that the size of the mobile anions has a large impact on chain packing in PILs. A well-packed structure is formed in the presence of larger mobile ions, while smaller ions frustrate the packing of PILs chains.

Very recently, Corvo et al. [51] studied the nanostructure of poly(1-vinyl-3-alkylimidazolium)s in the bulk, with varying alkyl side-chain lengths (*n* from *n* = 1 to *n* = 10) and counter-anions, using WAXS and SANS (small-angle neutron scattering). The WAXS patterns of bulk PIL features three peaks in the *Q* range 0.1 Å^−1^ < *Q* < 2.5 Å^−1^, as shown in Figure 4a. The low-*Q* peak (<0.5 Å^−1^) is assigned to the distance between two neighboring macromolecular chains, and when the alkyl chain length increases it sharpens, becomes more intense, and shifts to lower-*Q* values (higher distance between macromolecular chains). The intermediate peak shows a very small dependence on the alkyl side-chain length and is attributed to the correlation length of the counter-anion network. The higher-*Q* peak is ascribed to close contact between alkyl side chains inside the alkyl domain. Modelling of the SANS data, shown in Figure 4c,d, allowed the determination of the influence of the alkyl chain length on the radius of gyration *R_g_*, on the chain cross section and on the backbone-to-backbone distance. The backbone-to-backbone distance exhibited a non-monotonic variation with *n*, as side-chains tend to interpenetrate as their length increases.

SANS, supported by MD simulations and SAXS, has also been used in recent years to study the bulk-phase structure of ionic liquid mixtures [52,53,54]. Using the isotopic contrast variation technique, it was determined that the structure of these ionic liquid mixtures changes substantially as a function of composition. Evidence has been emerging in recent years that points to the existence of an enhanced level of structural complexity in IL-based systems [54], that results in hierarchical complex morphologies that play a role in the bulk properties of the systems (be it membranes, or others).

Despite all these advanced characterization studies performed on elucidating the nanostructure of PIL-based membranes for electrochemical applications, as well as elucidating the structure of ionic liquid mixtures, similar detailed morphological studies performed on PIL-based membranes for CO_2_ separation are still missing in the literature. As far as we know, neutron scattering techniques with contrast variation, such as small angle neutron scattering (SANS) and Spin-Echo-SANS (SESANS), have still not been used to study the micro and nanostructures of PIL-based membranes for CO_2_ separation. 

SANS probes the ~1–300 nm length scales, albeit in reciprocal space, whereas SESANS probes the 50 nm–15 μm length scale range [55,56]. Several studies have explored polymer systems such as colloids [57] and fibrous calcium caseinate gels [58]. However, it is only recently that new instrumentation at large facilities [59], and straightforward analysis techniques, have emerged for structural determination [60]. There also exist possibilities to use this technique to look at the kinetics [61], and a similar approach could be undertaken to look at the kinetics of film formation.

It is our belief that the design of PIL-based membranes for CO_2_ separation would greatly benefit from a better micro, nano, and molecular level understanding of micro- and nano-structure-performance relationships, which is still clearly missing. Relating the micro- and nano-structural features of the PIL-based membranes, to the understanding of their CO_2_ separation performances, remains therefore an open question up to now. Probing in more detail the structure of PIL-based membranes for CO_2_ separation will require the synergic use of advanced experimental (X-ray and neutron scattering) and computational tools.

## 4. Conclusions

PIL-based membranes have been studied for the separation of CO_2_ from gas mixtures. Among the different CO_2_ separations considered, namely CO_2_/CH_4_, CO_2_/H_2_, and CO_2_/N_2_, PIL-based membranes demonstrate more promising results in the separation of CO_2_ from N_2_. The best performing PIL-based membranes for CO_2_/N_2_ separation developed at laboratory scale in the last 5 years, exhibited a permeability of 542 barrer and a selectivity of 54, which is slightly above the 2008 Robeson upper bound.

The very high number of possible combinations of cations and anions creates the expectation of the large potential of this technology for CO_2_ separation. However, there seems to exist still a very large margin for further development. One particular aspect that requires much research improvement is related to the relationship between the microstructure and nanostructure of the PIL-based composite membranes, and their corresponding separation performances—this relationship is still poorly understood. In fact, studies regarding the relationship between PIL-based membranes’ nanostructure morphologies and their gas separation properties are still very scarce in the literature. The nanostructured morphologies are likely to impact on the transport properties, as well as on the mechanical properties of the membranes. To achieve this understanding, the micro- and nanostructures of the composite membranes should be studied with greater detail, using a battery of X-ray and neutron scattering techniques combined with theoretical ab-initio and molecular dynamics simulations. Although these advanced characterization techniques have been used in studies of PIL-based membranes for electrochemical applications, their use in the study of PIL-based membranes for CO_2_ separation has still not been reported.

## Figures and Tables

**Figure 1 polymers-15-00667-f001:**
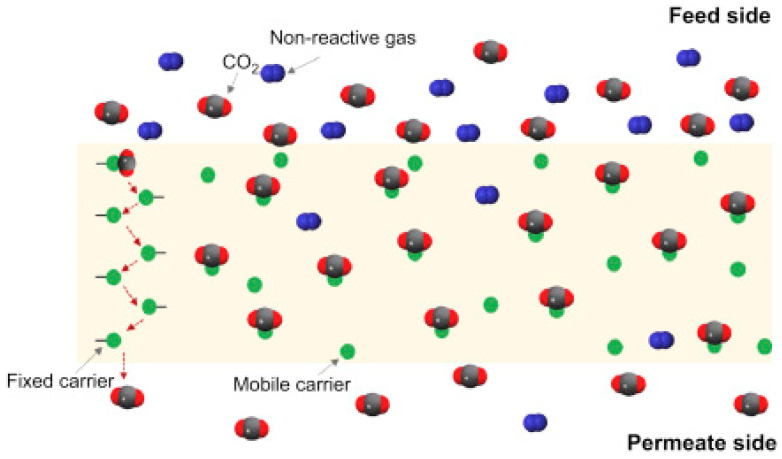
Schematic representation of the facilitated transport mechanism. Reproduced with permission from Ref. [27].

**Figure 2 polymers-15-00667-f002:**
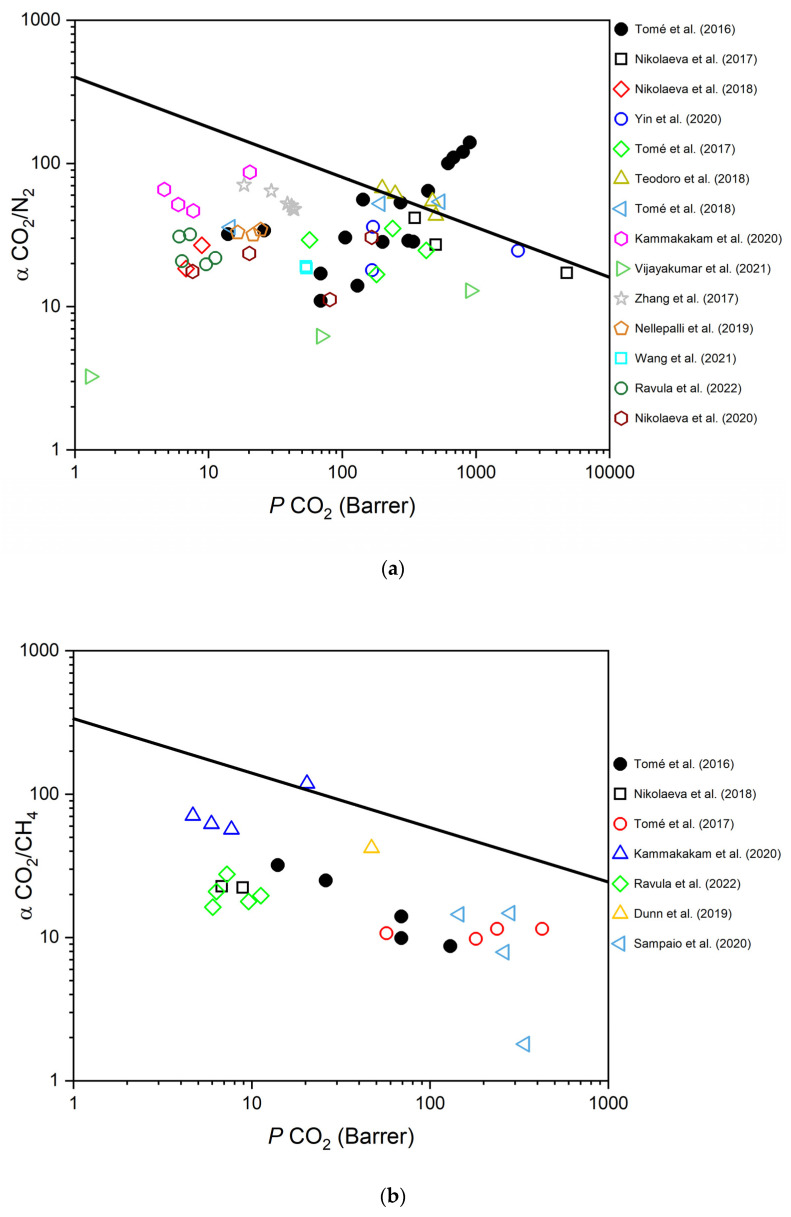
Robeson plots of PIL-based membranes for the: (**a**) CO_2_/N_2_ separation; (**b**) CO_2_/CH_4_ separation; and (**c**) CO_2_/H_2_ separation. The indicated works correspond to the following references: Tomé et al. (2016) [15]; Nikolaeva et al. (2017) [28]; Nikolaeva et al. (2018) [29]; Yin et al. (2020) [30]; Tomé et al. (2017) [31]; Teodoro et al. (2018) [32]; Tomé et al. (2018) [33]; Gouveia et al. (2018) [34]; Gouveia et al. (2021) [35]; Gouveia et al. (2020) [36]; Kammakakam et al. (2020) [37]; Vijayakumar et al. (2021) [38]; Zhang et al. (2017) [39]; Nellepalli et al. (2019) [40]; Wang et al. (2021) [41]; Ravula et al. (2022) [42]; Dunn et al. (2019) [43]; Nabais et al. (2019) [44]; Sampaio et al. (2020) [45] and Nikolaeva et al. (2020) [46].

**Figure 3 polymers-15-00667-f003:**
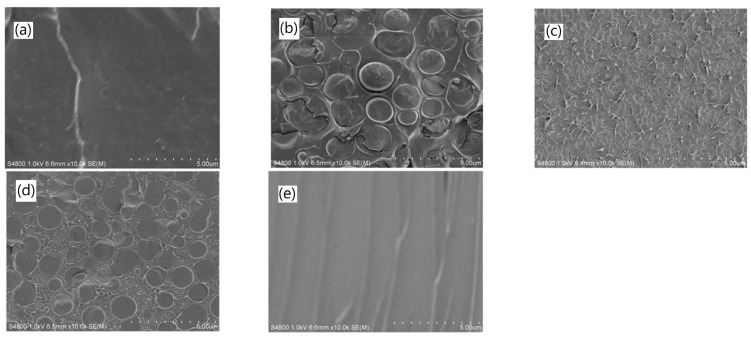
SEM images of the cross-section (×10,000) of the membranes: (**a**) pure PVAc, (**b**) PVAc/c-PIL-30, (**c**) PVAc/c-PIL-50, (**d**) PVAc/c-PIL-60, and (**e**) c-PIL. Structural features in the micrometer range are clearly visible [39].

**Figure 4 polymers-15-00667-f004:**
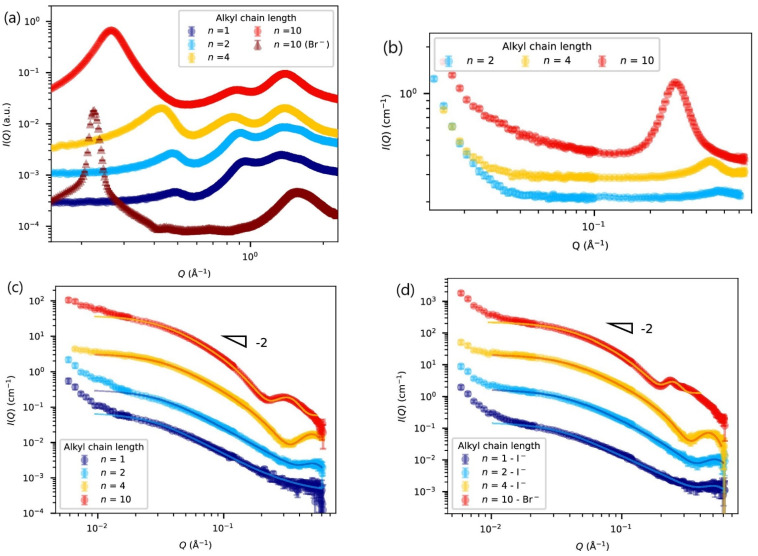
(**a**) WAXS data of bulk PCnTFSI (circle, open) with different alkyl side-chain lengths, *n*, and PC10Br (triangle, open), (**b**) SANS data of bulk purely deuterated PC*n*TFSI, (**c**) SANS data of bulk PC*n*TFSI H/D (1:1 *w*/*w*), and (**d**) SANS data of bulk PC*n*I and PC10Br H/D (1:1 *w*/*w*) with varying alkyl side-chain lengths, *n*. Solid lines correspond to the best model fits. All data in (**a**–**d**) are shifted vertically for clarity. All figures were readapted from Ref. [51].

**Table 1 polymers-15-00667-t001:** CO_2_ Separation Performance of Different Types of PIL-Based Membranes.

Membrane Type	Supported or Unsupported	Membrane Name	Temp (°C)	P CO_2_ (Barrer)	P N_2_	P H_2_	P CH_4_	*α* CO_2_/N_2_	*α* CO_2_/H_2_	*α* CO_2_/CH_4_	Year	Ref.
Neat PIL	S	P[VBTMA][Tf_2_N]	26	502	19	-----	-----	27.0 (a)	-----	-----	2017	[28]
		P[VBHEDMA][Tf_2_N]		349	8.3	-----	-----	41.6 (a)	-----	-----		
		P[VBMP][Tf_2_N]		4802	280	-----	-----	17.2 (a)	-----	-----		
	S	P[CA][Tf_2_N]	25	8.9	0.3	8.7	0.4	26.8	1.0	22.3	2018	[29]
		P[DADMA][Tf_2_N] (reference)		6.8	0.4	6.6	0.3	18.4	1.0	22.7		
	U	LP(1:2)	35	170	4.7	-----	-----	36	-----	-----	2020	[30]
		HP(1:2)		167	9.3	-----	-----	18	-----	-----		
PIL-IL composite membranes	U	PIL NTf_2_—40 IL Si NTf2	20	181	-----	-----	-----	16.8	-----	9.8	2017	[31]
		PIL NTf_2_—60 IL Si NTf2		426	-----	-----	-----	24.7	-----	11.5		
		PIL C(CN)_3_—40 IL Si C(CN)_3_		57	-----	-----	-----	29.3	-----	10.7		
		PIL C(CN)_3_—60 IL Si C(CN)_3_		238	-----	-----	-----	35.2	-----	11.5		
	U	PIL N(CN)_2_—60 IL C(CN)_3_	20	249.0	4.1	-----	-----	61.3	-----	-----	2018	[32]
		PIL C(CN)_3_—60 IL B(CN)_4_		472.7	8.7	-----	-----	54.4	-----	-----		
		PIL C(CN)_3_—40 IL N(CN)_2_		198.8	3.0	-----	-----	67.0	-----	-----		
		PIL B(CN)_4_—60 IL C(CN)_3_		502.1	11.6	-----	-----	43.1	-----	-----		
	U	Medium Mw PIL—20 IL	20	14.6	-----	-----	-----	35.9	-----	-----	2018	[33]
		Medium Mw PIL—40 IL		193	-----	-----	-----	52.3	-----	-----		
		Medium Mw PIL—60 IL		542	-----	-----	-----	54.0	-----	-----		
	U	PIL C(CN)_3_—40 [C_2_mim][C(CN)_3_]	20	139	-----	14.5	-----	-----	9.6	-----	2018	[34]
			35	209	-----	25.7	-----	-----	8.1	-----		
		PIL C(CN)_3_—60 [C_2_mim][C(CN)_3_]	20	438	-----	29.1	-----	-----	15.1	-----		
			35	505	-----	40.3	-----	-----	12.5	-----		
		PIL NTf_2_—40 [C_4_mpyr][NTf_2_]	20	119	-----	21.9	-----	-----	5.4	-----		
			35	164	-----	34.4	-----	-----	4.8	-----		
		PIL NTf_2_—60 [C_4_mpyr][NTf_2_]	20	232	-----	29.8	-----	-----	7.8	-----		
			35	288	-----	46.0	-----	-----	6.3	-----		
		PIL NTf_2_—40 [C_2_mim][NTf2]	20	214	-----	26.2	-----	-----	8.2	-----		
			35	287	-----	43.8	-----	-----	6.5	-----		
	U	PIL C(CN)_3_—40 [C_2_mim][C(CN)_3_]	35	129.7	-----	15.7	-----	-----	8.2 (a)	-----	2021	[35]
		PIL C(CN)_3_—60 [C_2_mim][C(CN)_3_]		324.7	-----	28.3	-----	-----	11.4 (a)	-----		
		PIL NTf_2_—40 [C_4_mpyr][NTf2]		118.9	-----	23.6	-----	-----	5.0 (a)	-----		
		PIL NTf_2_—60 [C_4_mpyr][NTf2]		254.2	-----	38.3	-----	-----	6.6 (a)	-----		
		PIL NTf_2_—40 [C_2_mim][NTf2]		201.6	-----	29.0	-----	-----	6.9 (a)	-----		
	U	PIL TFSAM—20 IL TFSAM	35	40	1.1	12.4	-----	-----	3.2	-----	2020	[36]
		PIL TFSAM—40 IL TFSAM		177	5.0	24.6	-----	-----	7.2	-----		
		PIL FSI—20 IL FSI		38	0.9	10.5	-----	-----	3.6	-----		
		PIL FSI—40 IL FSI		201	4.5	22.7	-----	-----	8.9	-----		
		PIL TSAC—20 IL TSAC		72	2.8	20.2	-----	-----	3.5	-----		
	U	MIL—CF_3_/PEGDA(20%)/IL(0.5 equiv)	20	7.69	0.165	2.83	0.136	46.6	2.71	56.54	2020	[37]
		MIL—CF_3_/PEGDA(20%)/IL(1 equiv)		5.94	0.115	1.87	0.096	51.65	3.18	61.88		
		MIL—C_7_H_7_/PEGDA(20%)/IL(0.5 equiv)		4.67	0.071	1.65	0.066	65.77	2.83	70.75		
		MIL—C_7_H_7_/PEGDA(20%)/IL(1 equiv)		20.4	0.235	4.97	0.172	86.81	4.1	118.6		
	U	HT—66 wt% IL		2070	84.1			24.6			2020	[30]
	U	ILPPO	25	69.58	11.20	-----	-----	6.21	-----	-----	2021	[38]
		ILPPO/Br-6-MPRD-2		907.20	70.10	-----	-----	12.94	-----	-----		
		ILPPO/Br-6-MPRD-5		1.30	0.40	-----	-----	3.25	-----	-----		
PIL-polymer blend membranes	U	PVAc/ c-PIL-50 semi-IPN	20	18.43	0.26	-----	-----	70.61	-----	-----	2017	[39]
			30	29.47	0.46	-----	-----	64.20	-----	-----		
			40	38.86	0.74	-----	-----	52.30	-----	-----		
			50	41.72	0.83	-----	-----	50.56	-----	-----		
			60	42.72	0.89	-----	-----	48.00	-----	-----		
			70	43.88	0.92	-----	-----	47.70	-----	-----		
Block-copolymer	U	poly(ViPenIm)(Sty)NTf_2_—10% IL	20	21.6	0.68	-----	-----	31.7	-----	-----	2019	[40]
		poly(ViBenIm)(Sty)NTf_2_—25% IL		16.5	0.50	-----	-----	32.9	-----	-----		
		poly(ViNapIm)(Sty)NTf_2_—30% IL		24.5	0.71	-----	-----	34.4	-----	-----		
	U	SiO_2_-g-PMMA-b-PIL		54.1	2.9			18.5			2021	[41]
		SiO_2_-g-PIL		53.4	2.8			19.1				
	U	HM	20	6.34	0.29	3.98	0.31	20.76	1.59	20.92	2022	[42]
		BCP1-C_4_		9.61	0.43	5.54	0.55	19.72	1.69	17.80		
		BCP2-C_4_		7.26	0.23	4.92	0.29	31.99	1.45	27.61		
		BCP1-C_6_		11.23	0.51	7.69	0.58	21.82	1.48	19.61		
		BCP2-C_6_		6.05	0.21	3.86	0.34	30.82	1.57	16.28		
Mixed Matrix Membrane (MMM)	S	1d/IL/zeolite (64/16/20)		47			1.1			42	2019	[43]
	U	PIL-IL	30	47.1	------	------	------	------	2.2	------	2019	[44]
		MMM/10% MIL-53		35.2	------	------	------	------	3.4	------		
		MMM/20% MIL-53		50.0	------	------	------	------	6.5	------		
		MMM/30% MIL-53		89.0	------	------	------	------	13.3	------		
		MMM/10% Cu_3_(BTC)_2_		40.0	------	------	------	------	2.6	------		
		MMM/20% Cu_3_(BTC)_2_		74.3	------	------	------	------	3.0	------		
		MMM/30% Cu_3_(BTC)_2_		77.1	------	------	------	------	6.4	------		
		MMM/10% ZIF-8		54.8	------	------	------	------	3.0	------		
		MMM/20% ZIF-8		83.8	------	------	------	------	3.6	------		
		MMM/30% ZIF-8		97.2	------	------	------	------	4.5	------		
	U	Tf_2_N/40 IL BETI		146			10			14.5	2020	[45]
		Tf_2_N/40 IL BETI/10 MOF-5		261			33			7.9		
		Tf_2_N/40 IL BETI/20 MOF-5		282			19			14.8		
		Tf_2_N/40 IL BETI/30 MOF-5		340			189			1.8		
	S	P9IL0Zn0	25	7.6	0.4	------	------	17.6	------	------	2020	[46]
		P9IL6Zn0		166	5.4	------	------	30.5	------	------		
		P9IL6Zn1		80.9	7.2	------	------	11.2	------	------		
		P9IL6Zn9		20.2	0.9	------	------	23.5	------	------		

(a) Mixed-gas selectivity.

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
