# Peer review of "Recent Advances in Poly(Ionic Liquid)-Based Membranes for CO2 Separation"

_polymers, 2023, doi:10.3390/polym15030667_

Round 1

Reviewer 1 Report

This review presents few different types of PIL-based membranes for CO2 separation, such as neat PIL membranes, PIL-IL composite membranes and PIL-polymer blend membranes. The state-of-the-art separation results of the PIL-based membranes in recent 5 years are summarised in term of several gas pairs (CO2/N2, CO2/H2 and CO2/CH4). However, this article is lack of novelty and impacts to the audience. A similar work has been published recently with more advanced and comprehensive discussion provided [Sep. Purif. Technol. 299 (2022) 121784]. The authors are advised to revise the manuscript and emphasis the importance and uniqueness of this review compared to the published work.

Author Response

Our Answer:

We thank the reviewer for these pertinent comments and for indicating us a related review published recently [Sep. Purif. Technol. 299 (2022) 121784] which we were not aware of when we first submitted this article last month. We have now added a citation to this recent review [Sep. Purif. Technol. 299 (2022) 121784] – (ref. 18). Concerning the comment that our article lacks novelty and is similar to this recent review [Sep. Purif. Technol. 299 (2022) 121784], we have read this review very carefully and we respectfully disagree with the reviewer. Contrary to the review [Sep. Purif. Technol. 299 (2022) 121784], our review places a special focus on the micro- and nano-morphological characterization of the PIL-membranes which in our opinion requires a deeper research study and understanding.

We do believe that the reviewer found lack of novelty in our review because we have not emphasized enough the structural and morphological characterization of this type of membranes. Therefore, we have made the following changes to this review:

  • We have re-written the last part of the abstract which now reads: “The micro- and nano-morphological characterization of the membranes is highlighted as a research topic that requires deeper research study and understanding. Nowadays there is an array of advanced structural characterization techniques, such as neutron scattering techniques with contrast variation (using selective deuteration), that can be used to probe the micro- and nanostructure of membranes in length scales ranging from ~ 1 nm to ~ 15 m Although some of these techniques have been used to study the morphology of PIL-based membranes for electrochemical applications, their use in the study of PIL-based membranes for CO2 separation is still unknown.
  • In section “3 – Advanced structural characterization of PIL-based membranes” we have added the following paragraph about SANS and SESANS: “SANS probes the ~1 – 300nm length scales albeit in reciprocal space whereas SESANS probes the 50 nm – 15micron length scale range [55, 56]. Several studies have explored polymer systems such as colloids [57], fibrous calcium caseinate gels [58]. However, it is only recently that new instrumentation has emerged at large facilities [59] and straightforward analysis techniques for structural determination [60]. There also exists possibilities for this technique looking at the kinetics [61] and a similar approach could be undertaken to look at the kinetics of film formation.”

Reviewer 2 Report

1.     The authors must clarify whether this article is a comprehensive or short review.

2.     The authors should put more keywords, at least five.

3.     In the introduction lines 26-31 and 65-82, the authors need to mention the reference. It is a review paper, so either author should mention that these lines are their thought or someone else.

4.     The authors should avoid lumped references, and extracting the information from each reference is recommended.

5.     Lines 151—152 belong to which study?

6.     For better understanding, the authors should provide figures and tables in their respective sections.

7.     The authors need to put more studies from the selected years.

8.     The current form of the article needs extensive formatting.

Author Response

Reviewer #2

  1. The authors must clarify whether this article is a comprehensive or short review.

Our Answer: On the 2nd line of the abstract it is clearly mentioned that it is a short review.

  1. The authors should put more keywords, at least five.

Our Answer: We have added two additional keywords (structural and morphological characterization; small-angle scattering techniques)

  1. In the introduction lines 26-31 and 65-82, the authors need to mention the reference. It is a review paper, so either author should mention that these lines are their thought or someone else.

Our Answer: We thank the reviewer for this pertinent comment. We have added several new references.

  1. The authors should avoid lumped references, and extracting the information from each reference is recommended.

Our Answer: We respectfully disagree with this comment. Lumped references are quite common in review articles and in this review there are few lumped references. It is way of providing the readers with several examples of one same thing.

  1. Lines 151—152 belong to which study?

Our Answer: We thank the reviewer for this comment. We have removed these two lines.

  1. For better understanding, the authors should provide figures and tables in their respective sections.

Our Answer: We have placed the figures in their respective sections. However, we have left the Table in the end because in our opinion placing the Table in the middle of the article will “break” the review in two parts. Anyway, we hope that the editorial team will advise us about this.

  1. The authors need to put more studies from the selected years.

Our Answer: We thank the reviewer for this comment. We have found and cited 3 additional studies – References 30, 43 and 45.

  1. The current form of the article needs extensive formatting.

Our Answer: We have placed the Figures in their sections. We believe that final formatting will be also a responsibility of the editorial team.

Reviewer 3 Report

Interesting work. I think it is better to explain symbols of equations in lines 92 and 109.

Lists of acronyms is omissed.

Author Response

Reviewer #3

Interesting work. I think it is better to explain symbols of equations in lines 92 and 109. Lists of acronyms is omissed.

Our Answer: We thank the reviewer for this comment. We have explained the symbols.

Round 2

Reviewer 1 Report

We would like to reject the article due to similar review that published recently with more comprehensive discussion has been published.

Reviewer 2 Report

1. The authors addressed all the comments satisfactorily. The paper can be accepted after a language check.